

# Immune activity score to assess the prognosis, immunotherapy and chemotherapy response in gastric cancer and experimental validation

Xuan Wu[1,2,3,*], Fengrui Zhou[1,2,3,*], Boran Cheng[1,2,3], Gangling Tong[1,2,3], Minhua Chen[4], Lirui He[5], Zhu Li[1,2,3], Shaokang Yu[1,2,3], Shubin Wang[1,2,3] and Liping Lin[6]

[1] Department of Medical Oncology, Peking University Shenzhen Hospital, Shenzhen, China
[2] Shenzhen Key Laboratory of Gastrointestinal Cancer Translational Research, Shenzhen, China
[3] Cancer Institute of Shenzhen-PKU-HKUST Medical Center, Shenzhen, China
[4] Community Healthcare Center of Shenzhen Traditional Chinese Medicine Hospital, Shenzhen, China
[5] Department of Gastrointestinal Surgery, Peking University Shenzhen Hospital, Shenzhen, China
[6] Department of Oncology, Panyu Central Hospital, Cancer Institute of Panyu, Guangzhou, China
[*] These authors contributed equally to this work.

Corresponding authors
Shubin Wang,
shubinwang2013@163.com
Liping Lin,
linliping@pyhospital.com.cn

## ABSTRACT

**Background**. Gastric cancer (GC) is an extremely heterogeneous malignancy with a complex tumor microenvironment (TME) that contributes to unsatisfactory prognosis.
**Methods**. The overall activity score for assessing the immune activity of GC patients was developed based on cancer immune cycle activity index in the Tracking Tumor Immunophenotype (TIP). Genes potentially affected by the overall activity score were screened using weighted gene co-expression network analysis (WGCNA). Based on the expression profile data of GC in The Cancer Genome Atlas (TCGA) database, COX analysis was applied to create an immune activity score (IAS). Differences in TME activity in the IAS groups were analyzed. We also evaluated the value of IAS in estimating immunotherapy and chemotherapy response based on immunotherapy cohort. Gene expression in IAS model and cell viability were determined by real-time reverse transcriptase-polymerase chain reaction (RT-qPCR) and Cell Counting Kit-8 (CCK-8) assay, respectively.
**Results**. WGCAN analysis screened 629 overall activity score-related genes, which were mainly associated with T cell response and B cell response. COX analysis identified AKAP5, CTLA4, LRRC8C, AOAH-IT1, NPC2, RGS1 and SLC2A3 as critical genes affecting the prognosis of GC, based on which the IAS was developed. Further RT-qPCR analysis data showed that the expression of AKAP5 and CTLA4 was downregulated, while that of LRRC8C, AOAH-IT1, NPC2, RGS1 and SLC2A3 was significantly elevated in GC cell lines. Inhibition of AKAP5 increased cell viability but siAOAH-IT1 promoted viability of GC cells. IAS demonstrated excellent robustness in predicting immunotherapy outcome and GC prognosis, with low-IAS patients having better prognosis and immunotherapy. In addition, resistance to Erlotinib, Rapamycin, MG-132, Cyclopamine, AZ628, and Sorafenib was reduced in patients with low IAS.
**Conclusion**. IAS was a reliable prognostic indicator. For GC patients, IAS showed excellent robustness in predicting GC prognosis, immune activity status, immunotherapy

response, and chemotherapeutic drug resistance. Our study provided novel insights into the prognostic assessment in GC.

## INTRODUCTION

Gastric cancer (GC) is a heterogeneous and invasive malignant tumor of the digestive system (*Smyth et al., 2020*), accounting for a high proportion of cancer incidence and mortality. Global cancer statistics for 2020 showed that GC caused 760,000 death cases, with a recent incidence rate of more than 1 million cases (*Sung et al., 2021*). Diagnostic and therapeutic techniques for GC have developed rapidly over the past few decades, but the prognosis for patients with GC remains extremely poor, with a 5-year survival rate of less than 20% for advanced patients (*Tan, 2019*). Numerous studies supported that even these patients have similar tumor staging and histological typing, the heterogeneity of GC can cause significant differences in patient survival outcomes (*Smyth et al., 2020*). Therefore, mining specific tumor prognostic biomarkers and developing specific clinical therapies to improve survival outcomes in GC patients has crucial significance.

Tumor cells, infiltrating immune cells, stromal cells, and cytokines constitute complex TME, which might be responsible for the heterogeneity of GC (*Kaymak et al., 2021*; *Smyth et al., 2020*). Immunotherapeutic approaches is a promising life-saving option for cancer patients (*Chalabi et al., 2020*; *Sheih et al., 2020*; *Wang et al., 2019*). Currently, the most frequently employed immunotherapy modalities are immune checkpoint inhibitors (ICI), immune checkpoint blockade (ICB), with anti-PD-1/PD-L1 and anti-CTLA-4 being the most common immunotherapy mechanisms (*Jahanafrooz et al., 2020*; *Saleh et al., 2020*). CTLA-4 is one of the immune checkpoints expressed on the surface of T cells, which negatively regulates T cell-mediated immune responses. Tumor cells suppress anti-tumor response of immune cells through high-expressed CTLA-4 (*Sadeghi Rad et al., 2021*). ICI therapy achieves anti-tumor response by unlocking the depleting effect of immune cells to release the inhibitory effect of immune checkpoints (*Schumacher & Schreiber, 2015*). CTLA-4 inhibitor-related drugs have been proven to be therapeutically effective in GC. For example, Tremelimumab could be safely used for the treatment of advanced GC (*Evrard et al., 2022*). In addition, some immunotherapeutic agents targeting other immune checkpoints also exhibited promising therapeutic effects on GC. Clinical studies conducted on immunotherapies for GC (*e.g.*, ATTRACTION-2, KEYNOTE-059) have shown varied results in objective remission rates that range from 10% to 26% (*Fuchs et al., 2018*; *Kang et al., 2017*). The results of clinical trials indicated that only a minority of patients could sustainably benefit from immunotherapy. Therefore, differentiating patients with potential benefit from immunotherapy for GC remains a priority.

Immunotherapy is one of the current options for cancer treatment, but not all cancer patients could benefit from it, which points to the need for an accurate prediction of

therapeutic efficacy of immunotherapy. TIP database can be used to evaluate tumor immune circulating activity for predicting the therapeutic effect of immunotherapy (*Xu et al., 2018*). Based on TIP studies, many cancer signatures were developed. *Wang et al. (2022)* built a ferroptosis-associated prognostic signature for hepatocellular carcinoma by TIP-associated genes. Prognostic gene signatures were also validated in TIP (*Chi et al., 2022*). Overall, TIP contributes to the identification of immunotherapy-related signatures in cancer. In this study, the overall activity score was developed by normalizing the immune cycle score of GC acquired from the TIP database to assess the TME activity of GC. Then, the sequencing data of GC from TCGA database were analyzed to determine the gene modules highly relevant to the overall activity score using WGCNA. The TME activity was assessed by COX analysis to establish the IAS of TME activity signature for predicting GC prognosis, immunotherapy, and chemotherapy.

## MATERIAL AND METHODS

### Dataset source and preprocessing

The sequencing data, somatic mutation data of GC patients in the training set were sourced from the TCGA database (TCGA-STAD; https://portal.gdc.cancer.gov/). Data of some patients with incomplete clinical information and somatic mutation data were preprocessed in the SangerBox database (http://www.sangerbox.com/home.html) (*Shen et al., 2022*). Patients with survival time >0 were retained, whereas those with incomplete pathological staging were removed. After preprocessing, 350 tumor specimens and 31 paracancer specimens with complete clinical information in TCGA-STAD remained. For somatic cell data analysis, samples with missing single nucleotide variants (SNV) data or copy number variation (CNV) data were removed, finally, we had 437 and 443 GC specimens with complete SNV and CNV data. In addition, the sequencing dataset of GC (GSE26942) was extracted from the Gene Expression Omnibus (GEO) database (https://www.ncbi.nlm.nih.gov/geo/) as a validation set and then processed according to the above criteria. Finally, 93 GC specimens with complete clinical information remained.

### Overall activity score

The tumor immune cycle activity of GC tumors was evaluated by integrating the 7-step immune scores in the TIP database, and then the overall activity score was determined by normalizing the 7-step immune scores. Differences in overall activity scores were compared between tumor specimens, paracancer specimens and pathological stage groups using the Wilcox test ($p < 0.05$).

### Immuno-infiltration analysis

Immune cell infiltration analysis was performed using ESTIMATE (*Yoshihara et al., 2013*) and CIBERSORT algorithms (*Chen et al., 2018*). A total of 28 immune cell signature genes were collected from previous research and their activity scores for characterizing TME activity were evaluated by the single sample gene set enrichment analysis (ssGSEA) method (*Barbie et al., 2009*; *Charoentong et al., 2017*). In addition, immune-related pathways from the Kyoto Encyclopedia of Genes and Genomes (KEGG; https://www.kegg.jp/) database were subjected to the ssGSEA method to assess pathway activities.

## WGCNA

To further screen genes relevant to the overall activity score, the limma package (*Ritchie et al., 2015*) was used for differential analysis between tumor specimens and paracancer specimens (|log2 fold change| > 1, FDR < 0.05) to obtain differentially expressed genes (DEGs) in tumor specimens. Then WGCNA was performed based on the DEGs (*Langfelder & Horvath, 2008*) with the parameters of height = 0.15 and deepSplit = 3. Gene modules sharing a high similarity in the network were merged into a new one. The overall activity score was considered as traits for Pearson correlation analyses with the eigengenes characterizing each module to access the relevance of gene modules to the overall activity score. Biological functions were analyzed using Gene Ontology (GO) and KEGG analyses in the WebGestaltR (*Liao et al., 2019*) package.

## Construction of IAS

In TCGA-STAD, univariate COX analysis was performed to identify prognosis-related genes from the module genes for GC. Then the most significant genes affecting the prognosis of GC were determined by Least absolute shrinkage and selection operator (LASSO) and multivariate COX analysis. IAS was constructed based on regression coefficients and expression for each gene. Samples were divided into high IAS and low IAS groups based on the grouping threshold of IAS = 0. Kaplan–Meier survival analysis and ROC analysis were conducted in the timeROC package to assess the prognostic guidance value of IAS (*Blanche, Dartigues & Jacqmin-Gadda, 2013*). The robustness of IAS was validated in the validation set GSE26942.

## Mutation analysis

The mutation landscape of patients in the IAS groups was analyzed. Firstly, we calculated the tumor mutation burden (TMB) for each patient in the two IAS groups and compared the spearman correlation among IAS, TMB and overall activity score. For somatic mutation data, SNV mutation and CNV data were evaluated using the maftools package (*Mayakonda et al., 2018*), and their high-frequency mutation sites were evaluated by the gisticOncoPlot function and waterfall plots were generated.

## Evaluation of immunotherapy

Tumor immune dysfunction and exclusion (TIDE) scores from the TIDE database (http://tide.dfci.harvard.edu/) were collected for assessing the risk of immune escape (*Jiang et al., 2018*) so as to assess the value of IAS as a guiding tool for immunotherapy response. Next, sequencing data and clinical information of GC patients treated with an-PD-L1 drug (atezolizumab) were retrieved from the IMvigor210 cohort (http://research-pub.gene.com/IMvigor210CoreBiologies/). Based on clinical information, patients in the IMvigor210 cohort were classified as stable disease (SD), complete response (CR), partial response (PR), and progressive disease (PD). In the four cohorts of patients, the value of IAS as a clinical guide to immunotherapy was explored.

## Chemotherapy drug sensitivity analysis

The expression profile data of GC cells treated with the chemotherapeutic drugs (Erlotinib, Rapamycin, MG-132, Cyclopamine, AZ628, and Sorafenib) retrieved from the Genomics

of Drug Sensitivity in Cancer database (GDSC; https://www.cancerrxgene.org/), and the half maximal inhibitory concentration (IC50) of these drugs was determined by the pRRophetic package. The IC50 values of the drugs were obtained by calculating the gene expression matrix of the samples in TCGA-STAD and performing ridge regression analysis using the linearRidge () function of the ridge package (*Geeleher, Cox & Huang, 2014*). Immune checkpoint genes were extracted from the study of *Hu et al. (2021)* and their levels were evaluated in IAS subgroups.

## PPI network

We obtained immune-related genes from the pan-cancer analysis of *Charoentong et al. (2017)*. We calculated the expression correlation between AKAP5, CTLA4, LRRC8C, AOAH-IT1, NPC2, RGS1, SLC2A3 and immune-related genes and selected a total of 145 genes with abs(cor) > 0.4 to develop a protein-protein interaction network in STRING database (https://string-db.org/). Finally, five key genes were contained in the risk model, namely, RGS1, AKAP5, SLC2A3, CTLA4, and LRRC8C.

## Nomogram analysis

In TCGA-STAD, univariate and multivariate COX analyses were performed by integrating Age, Stage, and IAS to determine the clinical factors affecting GC prognosis and to construct a nomogram. Further, the clinical efficacy of nomogram and IAS was assessed by plotting the decision curve.

## Cell culture and transient transfection

Beina Biotechnology Institute (China) provided the human GC cell lines HGC-27, AGS and the normal epithelial cells of human gastric mucosa RGM-1. F12 DMEM medium containing 10% fetal bovine serum was used for cell culture. All the cell lines were maintained at 37 ° C and 5% $CO_2$ in a humid incubator.

AKAP5 siRNA (Sigma, China) and AOAH-IT1 siRNA (Sigma, China) was transfected into the cells applying Lipofectamine 2000 (Invitrogen, Waltham, MA, USA). The target sequences for PPARG siRNAs were AACCACAATTTCAGAAATTCATG (AKAP5-si) and ATCATGAGTAGGTTAGACATTTA (AOAH-IT1-si).

## RT-qPCR

Using the Trizol reagent (Sigma-Aldrich, St. Louis, MO, USA), total RNA was separated from RGM-1, HGC-27, and AGS, respectively. RT-qPCR was performed with 2 μg RNA in each sample using FastStart SYBR Green Master (Roche, South San Francisco, CA, USA) and ABI Q5 PCR System (Roche, South San Francisco, CA, USA). cDNA together with 2 μl of cDNA template, 0.5 ul of forward and reverse primers, and water in a required amount of 20 μl served as a template. The PCR reactions were operated under the cycling conditions that began with DNA denaturation for 30 s at 95 ° C, followed by 45 cycles for 15 s at 94 ° C, for 30 s at 56 ° C, and for 20 s at 72 ° C. See Table 1 for the sequence list of primer pairs of the target genes.

**Table 1  The primers of genes.**

| Gene | Forward primer sequence (5–3) | Reverse primer sequence (5–3) |
| --- | --- | --- |
| AKAP5 | GCCATTGGAGGGTGAAATGC | CCTTTTTGGCCCTCTTGGGA |
| CTLA4 | GCCCTGCACTCTCCTGTTTTT | GGTTGCCGCACAGACTTCA |
| LRRC8C | GGGATGTGTTTACCGATTACCTC | CTGCACTCTTTTCGGAAGGC |
| AOAH-IT1 | GACCCATGGTTCCAACGCTA | CGTCTGGCTCTGGGAGATTC |
| NPC2 | TCCTGGCAGCTACATTCCTG | ACAGAACCGCAGTCCTTGAAC |
| RGS1 | TCTTCTCTGCTAACCCAAAGGA | TGCTTTACAGGGCAAAAGATCAG |
| SLC2A3 | GCTGGGCATCGTTGTTGGA | GCACTTTGTAGGATAGCAGGAAG |
| ACTB | CATGTACGTTGCTATCCAGGC | CTCCTTAATGTCACGCACGAT |

## Cell viability

Following the manufacturer's protocol, CCK-8 (Beyotime, Beijing, China) was performed to analyze the cell viability. Various cells with designed treatments were cultured at a density of $1 \times 10^3$ cells per well in 96-well plates. CCK-8 solution was added to the cells at indicated time points. We used a microplate reader to detect the O.D 450 value of each well after 2-h incubation at 37 ° C.

## Transwell assay

Invasion of HGC-27 and AGS cell lines were detected by performing Transwell assays. The cells ($5 \times 104$) were inoculated into Matrigel-coated chambers (BD Biosciences, San Jose, CA, USA). Complete DMEM medium was added to the lower layer and serum-free medium was added to the upper layer. Migrating or invading cells were fixed with 4% paraformaldehyde after 24-h h incubation and then dyed by 0.1% crystalline violet. Cell counting was performed under a light microscope.

## Statistical analysis

In this study, the analysis and plotting of sequencing data were all based on R software (version: 3.6.1). Experimental data statistics were done using Graphpad Prism 8 Software (GraphPad, San Diego, CA, USA). In the results, ns represented no significance, meaning $p > 0.05$. * represented $p < 0.05$, ** represented $p < 0.01$, *** represented $p < 0.001$, and ****p represented $P < 0.00001$.(X) represented correlation coefficient $r < 0.2$ in the correlation analysis, meaning a weak correlation.

# RESULTS

## Overall activity score in GC

This study analyzed the expression pattern of 7-step signature genes in the cancer immune cycle in GC specimens and paracancerous tissue specimens from the TCGA-STAD cohort. From Fig. 1A, it could be observed that 7-step signature genes were activated in tumor specimens. Overall activity scores were higher in tumor specimens compared to paraneoplastic specimens (Fig. 1B). Overall activity score in tumor tissues increased with a higher staging compared to tumor specimens with low clinicopathological staging (Figs. 1C–1E).

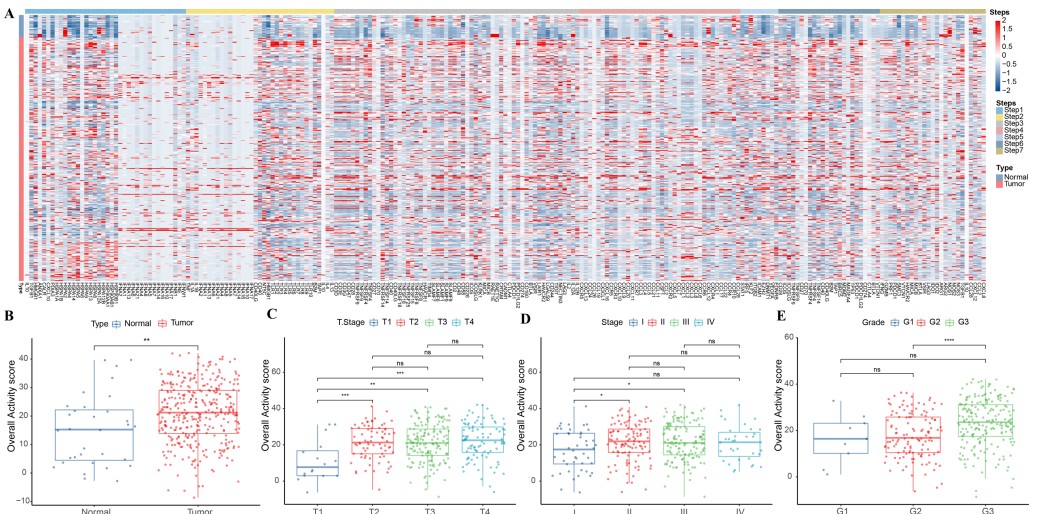

**Figure 1** **Overall activity score in gastric cancer.** (A) Heatmap of 7-step signature gene expression in tumor specimens and paracancer specimens. (B) Overall activity score in the normal and paraneoplastic specimen groups. (C–E) Overall activity score in T Stage, Stage, and Grade groups.

## Association between overall activity score and TME score

The correlation between overall activity score and ESTIMATEScore ($R = 0.79$, $p < 2.2e-16$), ImmuneScore ($R = 0.86$, $p < 2.2e-16$), StromalScore ($R = 0.6$, $p < 2.2e-16$) was positive (Fig. 2A). The immune infiltration scores of 22 immune cells in TME were calculated by CIBERSORT, and we found that macrophages M1 ($R = 0.35$), T cells CD4 memory activated ($R = 0.43$), T cells CD4 memory resting ($R = -0.42$), and T cells CD8 ($R = 0.4$) were closely correlated with overall activity score ($p < 0.05$) (Fig. 2B). In addition, the immune activity of 28-immune cells and 15 immune pathways in the TME of GC were measured by the ssGSEA method and we observed a remarkable positive correlation with overall activity score. Moreover, the majority of the immune cell activity of 28-immune cells and the majority of the 15 immune pathways were positively correlated (Fig. 2C).

## Identification of overall activity score-associated genes

In tumor specimens, 15,147 genes were filtered by limma package following differential analysis (FDR $< 0.05$). The expression profiles of 15,147 genes in 350 tumor specimens were further exploited to develop the WGCNA network with a scale-free $R^2$ of exactly 0.85 at a soft threshold $\beta = 10$, which met the scale-free network criteria (Figs. 3A–3B). Ten different patterns of co-expressed gene modules (height = 0.15, deepSplit = 3, minimum number of genes in the module >80) were identified based on the adjacency matrix and dynamic shearing algorithm (Fig. 3C). The eigenvector values (eigengenes) of the 10-gene modules as clinical features were correlated with the overall activity score. Pearson correlation analysis was then performed to identify overall activity score-associated genes in GC. Yellow ($R = 0.71$, $p = 8e-60$, number of genes: 376) and red ($R = 0.71$, $p = 2e-60$, number of genes: 253) modules were the two most significant gene modules associated with the overall activity score (Fig. 3D). The functions of 629 genes were assessed by GO

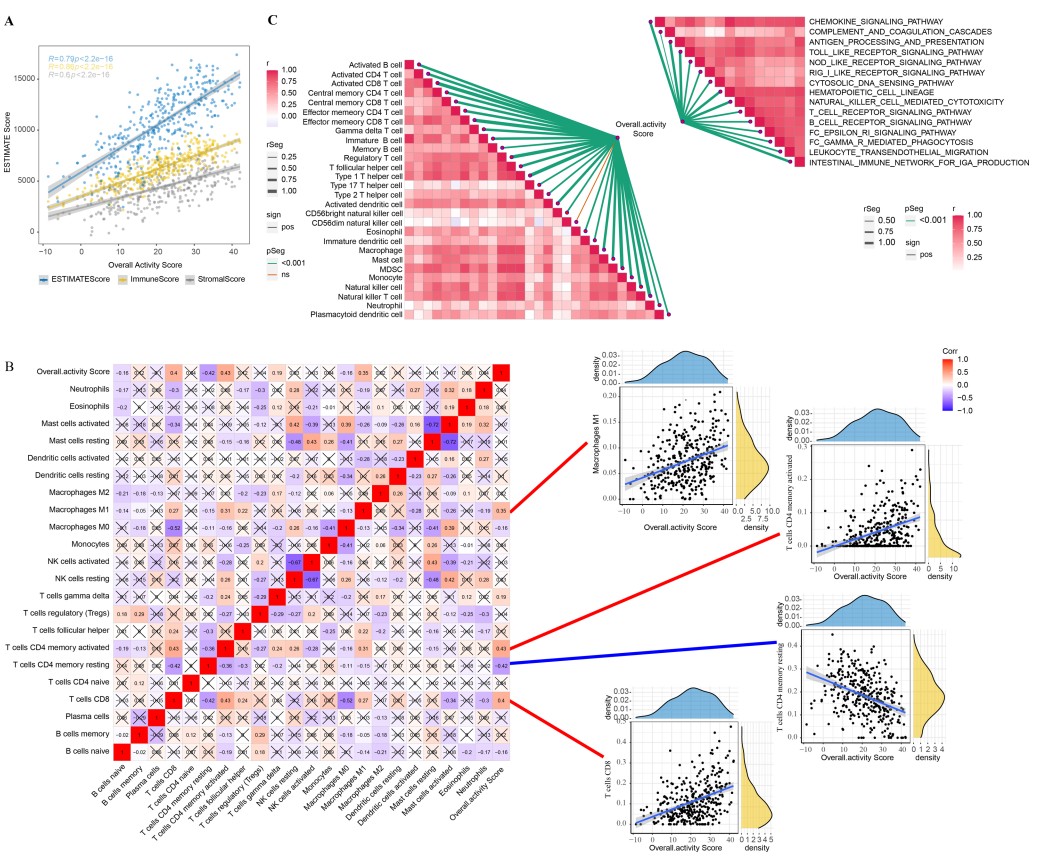

**Figure 2 TME activity in gastric cancer.** (A–C) ESTIMATE results, CIBERSORT results, immune-related pathways with Spearman correlation of overall activity score.

and KEGG enrichment analysis. These genes were enriched in 51 KEGG pathways and 900 GO terms (BP: 755, CC: 76, MF: 69), containing T cell responses, B cell responses. The top 10 prominent KEGG pathways, GO_BP terms, GO_CC terms, and GO_MF terms were displayed in Fig. 3E.

## IAS

In TCGA-STAD, 25 genes with prognostic relevance in GC ($p < 0.05$) in the yellow and red modules were screened by univariate COX analysis (Fig. 4A). When the penalty parameter lambda = 0.0189, genes with high similarity in the univariate COX were excluded, leaving a total of 16 significant genes (Figs. 4B–4C). Subsequently, seven genes (AKAP5, CTLA4, LRRC8C, AOAH-IT1, NPC2, RGS1, and SLC2A3) were determined by multivariate COX analysis as the most prognostically relevant genes for GC and composed together as a 7-gene signature for characterizing the immune activity of GC. The immune activity score (IAS $= -1.039 * AKAP5 - 0.36 * CTLA4 + 0.372 * LRRC8C + 1.037 * AOAH - IT1 + 0.364 * NPC2 + 0.226 * RGS1 + 0.135 * SLC2A3$) was developed. The IAS of all GC specimens were calculated by the formula and normalized by zscore. Samples with IAS > 0 were defined as the high IAS group ($n = 187$), while samples with IAS < 0 were defined as the low IAS

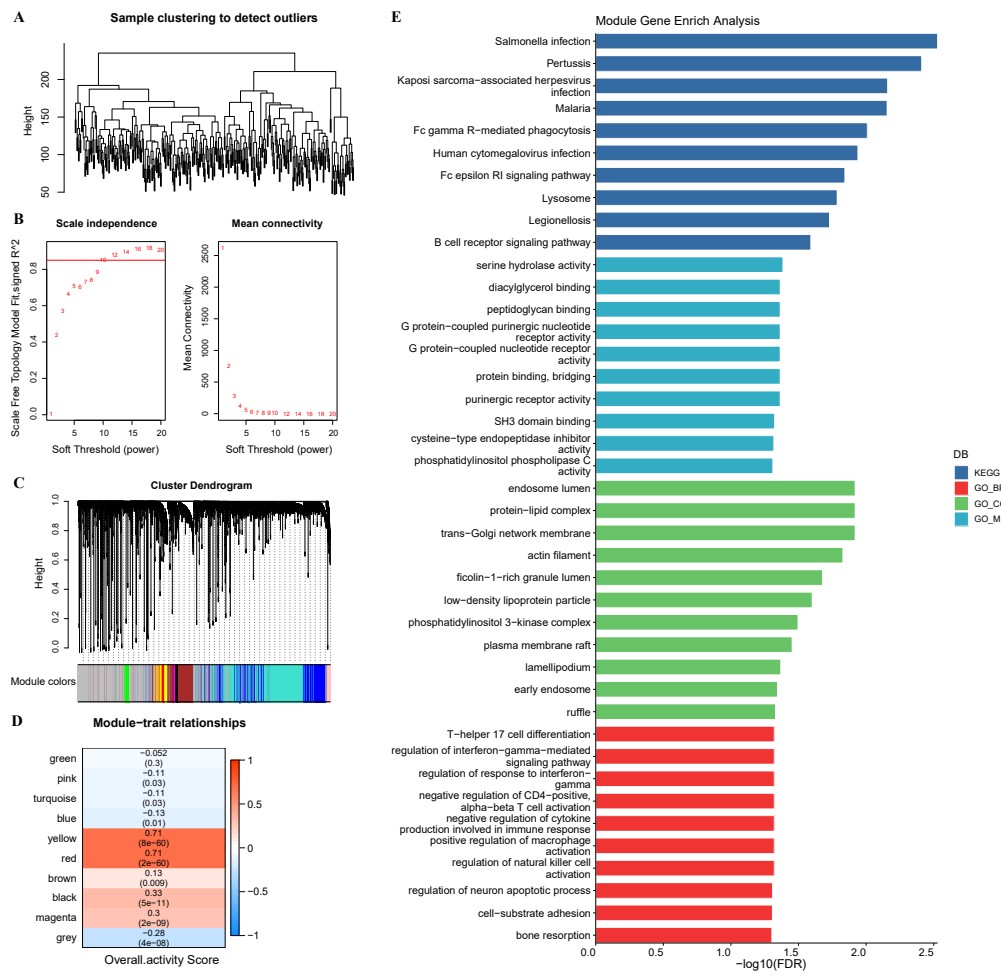

**Figure 3  WGCNA.** (A) Sample clustering tree. (B) Construction of scale-free network. (C) Gene modules (D) Trait correlation heat map. (E) Bar graph of GO, KEGG results.

group ($n = 163$). According to the scatter plot of the sample survival status distribution, GC patients with low IAS had a longer survival. The expression pattern of 7-gene was shown in Fig. 4D. ROC curves demonstrated that IAS showed excellent predictive performance in assessing GC prognosis (AUC = 0.7, 0.72, 0.79 at 1, 3 and 5 year(s), respectively) (Fig. 4E). Kaplan–Meier curves showed that five-year survival rate and median survival of patients in the high IAS group were reduced compared to those in the low IAS group (Fig. 4F), and this phenomenon was further validated in the validation set (GSE26942) (Figs. 4G–4H).

In addition, these seven genes were subjected to qRT-PCR and we found that the expression of AKAP5 and CTLA4 was downregulated in GC cell lines but upregulated in normal epithelium of gastric mucosa (Figs. 5A–5B). However, compared to normal gastric mucosal epithelial cells, LRRC8C, AOAH-IT1, NPC2, RGS1 and SLC2A3 were significantly upregulated in GC cell lines (Figs. 5C–5G). Subsequently, the expression of AKAP5 and AOAH-IT1 was inhibited with small interfering RNA in the two GC cell lines HGC-27 and

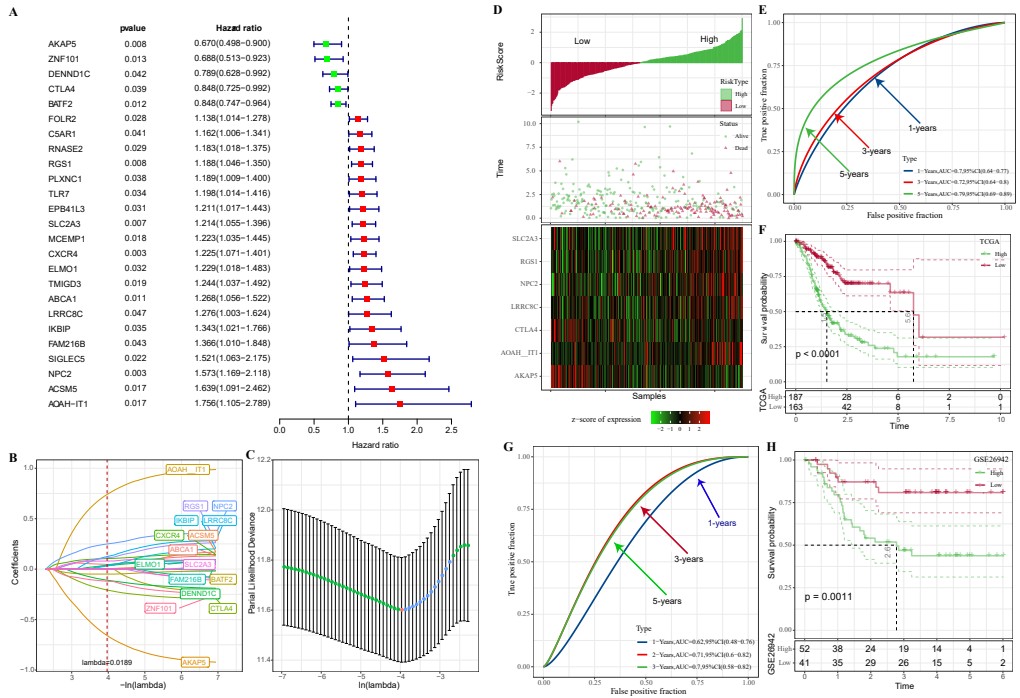

**Figure 4 Construction of IAS.** (A) Forest plot of univariate COX results. (B) Lambda change trajectory. (C) lambda selection interval (D) sample groups. (E) ROC curves. (F) Kaplan–Meier curves. (G) ROC curves for the GSE26942 dataset. (H) Kaplan–Meier curves for the GSE26942 dataset.

AGS. It could be observed that the viability of the two cells was increased after inhibition of AKAP5 (Figs. 5H–5I) but reduced after inhibition of AOAH-IT1 (Figs. 5J–5K). We subsequently examined alterations in cell migration and invasive capacity after inhibiting AKAP5 and AOAH-IT1 expression in HGC-27 and AGS cell lines. The results showed that the migration and invasion of HGC-27 and AGS cell lines were enhanced. Meanwhile, the migratory and invasive abilities of HGC-27 and AGS cell lines were significantly reduced after inhibition of AOAH-IT1 expression (Figs. 6A–6D).

## Correlation of IAS with clinical features and mutational features

IAS presented negative correlation with TMB (R = −0.21, $p = 8.9e-05$) (Fig. 7A). The top 15 CNV phenomenon appeared most frequently in the IAS groups with the same trend. Specifically, deletion was detected in AP_22:8q24.21, AP_52:20q13.2, AP_53:20q13.32, AP_51:20q13.12, AP_21:8q22.2, AP_20:8q21.13, AP_13:7p22.1, AP_14:7p11.2, AP_15:7q21.2 sites amplified, and DP_20:9p23, DP_10:4q34.3, DP_21:9p23, DP_29:16q23.1, DP_8:4q22.1, and DP_32:17p12 sites (Fig. 7B). In addition, we found that 15 genes in the distinct IAS groups with mutation frequencies higher than 5% (Fig. 7C). The correlation among IAS and clinical features and mutation features was analyzed accordingly. Firstly, patients with T2-3 ($p < 0.05$) and high stage (Stage IV, $p < 0.05$) showed higher IAS in comparison to T1 and Stage 1 patients, but this not distinct in the Grade groups. IAS demonstrated a positive correlation trend with overall activity score

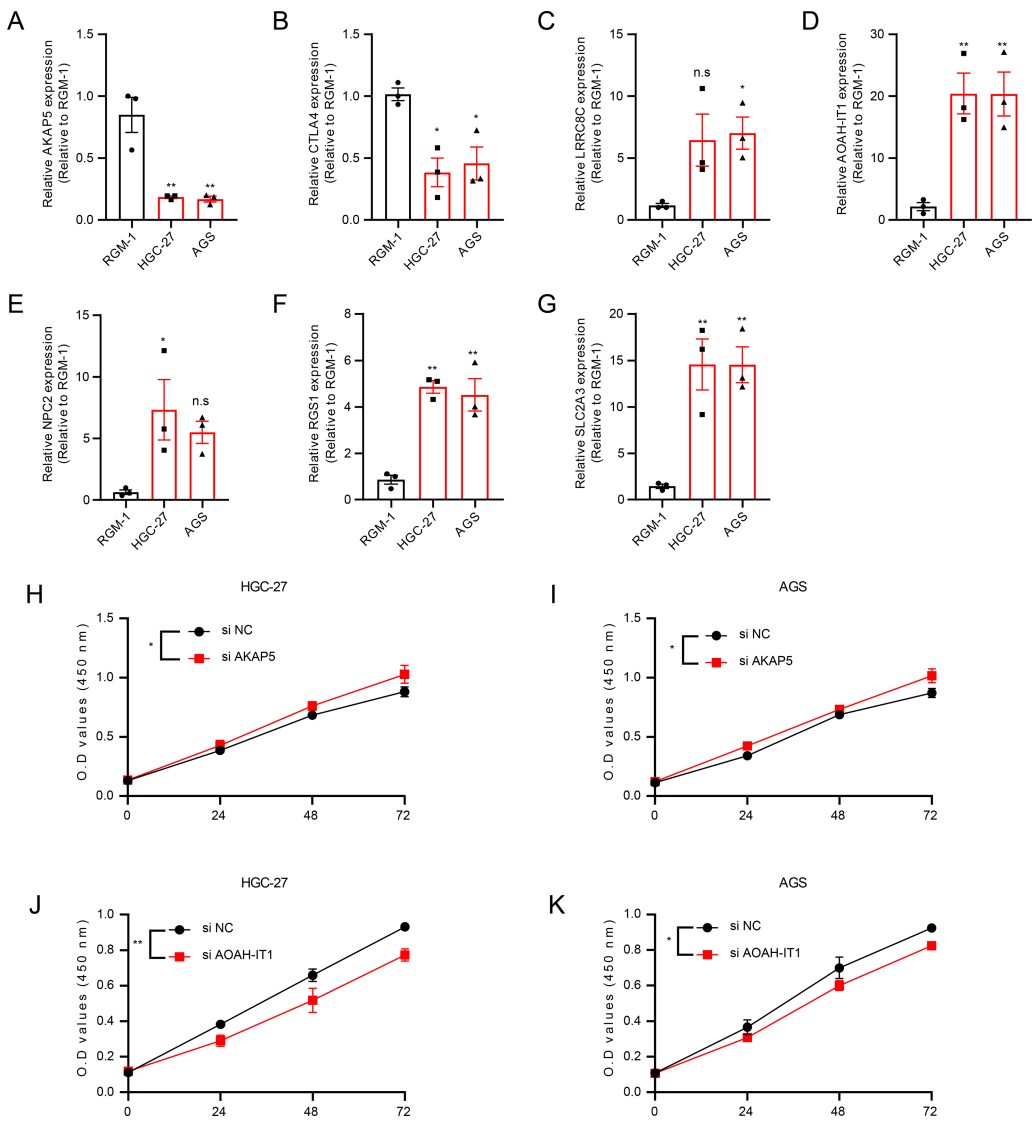

**Figure 5** **The validation of seven genes using experiments.** (A–G) mRNA Expression of AKAP5, CTLA4, LRRC8C, AOAH-IT1, NPC2, RGS1 and SLC2A3 in RGM-1, HGC-27 and AGS cells. (H–I) Cell viability of HGC-27 and AGS after inhibition of AKAP5 expression. (J–K) Cell viability of HGC-27 and AGS after inhibition of AOAH-IT1 expression. n.s > 0.05, * ≤ 0.05, ** ≤ 0.01. The results are presented as mean ± S.E.M. $n = 3$/group.

of GC patients ($R = 0.17$, $p = 0.0016$) (Fig. 7D). Comparison on the differences in CNV loci and mutated genes in the high and low IAS groups showed a higher proportion of AP_22:8q24.21, AP_21:8q22.2 mutations and a higher frequency of mutations in MUC16, ZFHX4 in the low IAS group (Figs. S1A–S1B).

## Association among IAS and TME scores and GC treatment

The ESTIMATE results demonstrated that patients in the high IAS group presented markedly higher StromalScore, ImmuneScore, ESTIMATEScore compared to the low IAS

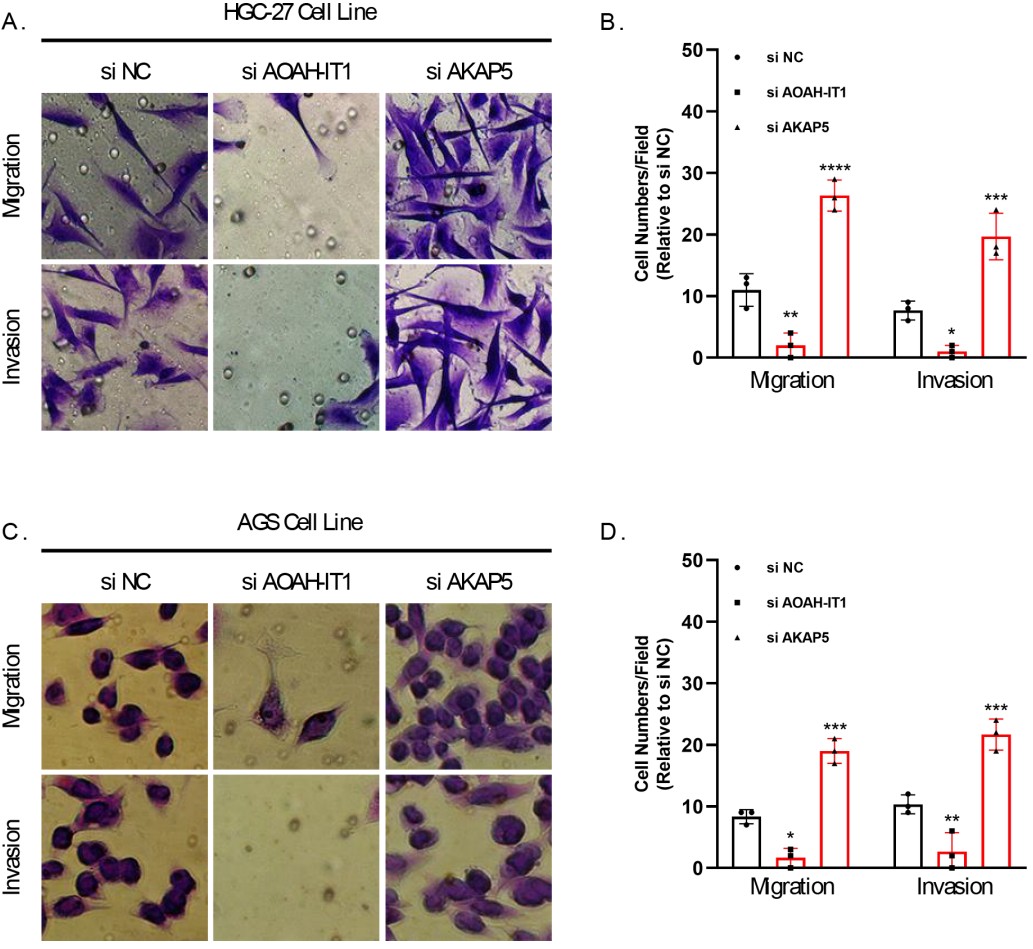

**Figure 6  Transwell assay for viability of HGC-27 and AGS cell lines after inhibition of AKAP5 and AOAH-IT1 expression.** (A–B) Altered migration and invasion ability of HGC-27 cell line after inhibition of AKAP5 and AOAH-IT1 expression. (C–D) Altered migration and invasion ability of AGS cell line after inhibition of AKAP5 and AOAH-IT1 expression. $N = 3$, * ≤0.05, ** ≤0.01, *** ≤0.001, **** ≤0.0001. The results are presented as mean ± SD.

group patients ($p < 0.05$) (Fig. 8A). Notably, IAS demonstrated positive correlation trend with StromalScore ($R = 0.42$, $p < 2.2e−16$), ImmuneScore ($R = 0.22$, $p = 4.8e−05$), ESTIMATEScore ($R = 0.35$, $p = 2.5e−11$) (Fig. 8B). IAS was positively related to immunoreactivity of 19 immune cells (Fig. 8C). From the TIDE analysis, we found that tumor cells in the TME of the high IAS group had a greater chance to escape from immune cell killing and immune escape may occur due to a higher TIDE score, which also meant that the high IAS group was probably not suitable for taking ICB treatment (Fig. 8D). Moreover, we identified immune-related genes with co-expression phenomena with the seven prognostic genes and generated a PPI network (Fig. S2).

The IAS of 348 samples from the IMvigor210 cohort treated with an-PD-L1 drug (atezolizumab) was also assessed. The IAS was higher in SD/PD (Fig. 9A) and the proportion of CR/PD was higher in the low IAS group (Fig. 9B). The Kaplan–Meier curves revealed

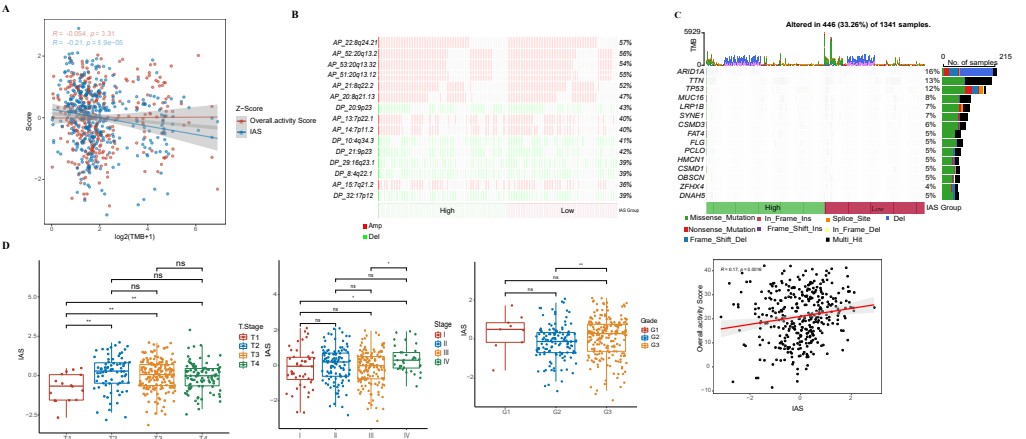

**Figure 7** **Correlation of IAS with clinical features and mutational features.** (A) Spearman correlation between IAS and TMB. (B) Heat map describing the frequency of CNV events between high-risk and low-risk groups. (C) The top 15 genes with the highest mutation frequency between the high-risk and low-risk groups. (D) IAS statistics in pathological groups.

that the low IAS group had longer median survival time and overall survival (Fig. 9C), indicating that IAS was an excellent assessment tool. Finally, the correlation between the tolerance to chemotherapeutic agents in distinct IAS groups was also evaluated. Compared to the high IAS group, the IC50 for Erlotinib ($p < 0.01$), Rapamycin ($p < 0.0001$), MG-132 ($p < 0.0001$), Cyclopamine ($p < 0.01$), AZ628 ($p < 0.001$), and Sorafenib ($p < 0.0001$) was higher (Fig. 9D). And according to this finding, patients with low IAS were less likely to acquire drug resistance to these six treatments than those with high IAS. We found that most of the activated and inhibited immune checkpoints showed higher expression levels in high IAS (Figs. S3A–S3B).

## GSEA and ssGSEA

According to the GSEA results, 14 HALLMARK pathways in the MsigDB database were markedly enriched in the high IAS group, mainly containing INFLAMMA-TORY_RESPONSE, IL6_JAK_STAT3_SIGNALING, ANGIOGENESIS, KRAS_SIGNALING_UP, IL2_STAT5_SIGNALING, these pathways were closely involved in the inflammatory immune response (Fig. 10A). In contrast, according to the results of ssGSEA, 39 pathways in the KEGG database showed increased activity, specifically, the activity of these pathways tended to increase with increasing IAS (Fig. 10B). Additionally, 19 of the 39 pathways were positively correlated with IAS ($p < 0.05$) and 2 pathways were negatively correlated with IAS ($p < 0.05$). The downregulated pathways were mainly associated with energy metabolic activities in cells, and the upregulated pathways were mainly associated with amino acid metabolic functions (Fig. 10C). Further, the spearman correlation between IAS and six inflammatory immune response pathways revealed that IAS mainly affected B cell receptor signaling pathway ($R = 0.11$, $p = 0.0392$), inflammatory response ($R = 0.329$,

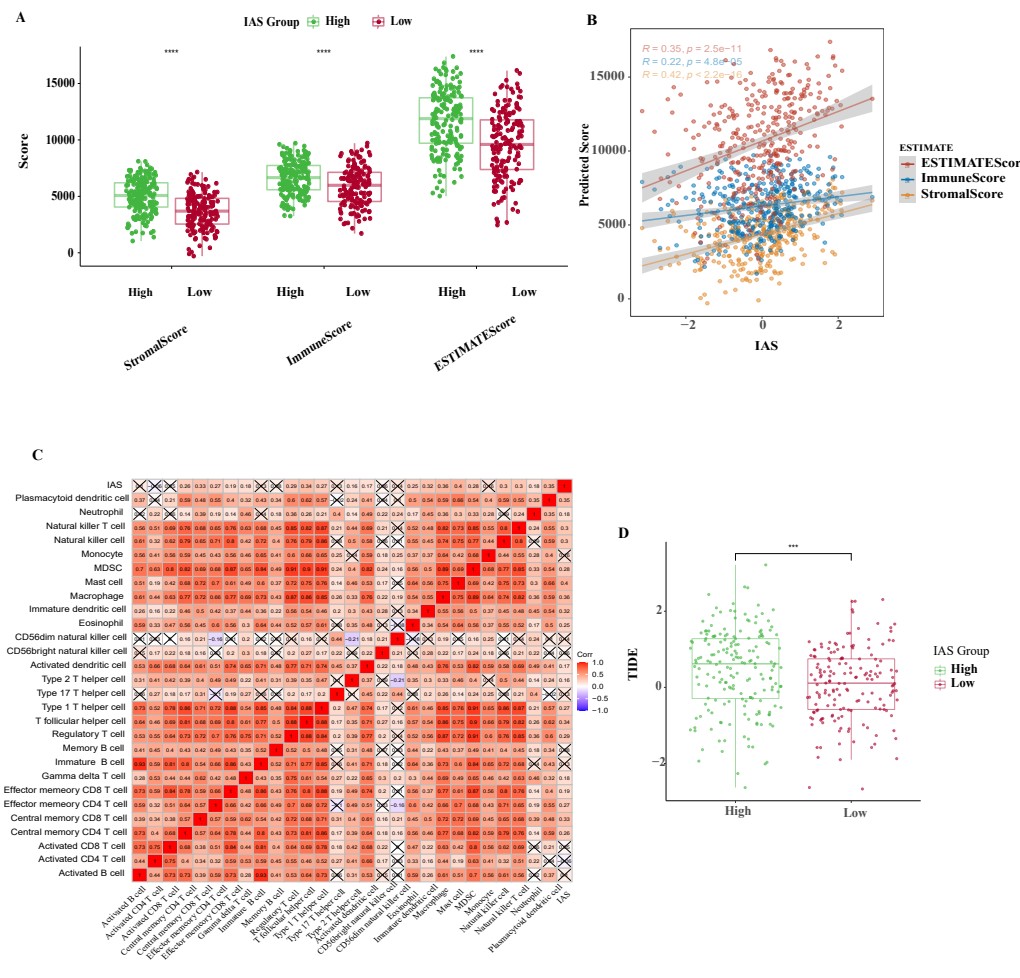

**Figure 8  TME activity in IAS groups.** (A) ESTIMATE results. (B) Spearman correlation of ESTIMATE results with IAS. (C) Spearman correlation of ssGSEA results with IAS. (D) TIDE scores.

$P = 3.54\text{e}{-}10$), Th1 and Th2 cell differentiation ($R = 0.116$, $P = 0.0299$), and Th17 cell differentiation ($R = 0.152$, $P = 0.00442$) (Fig. 10D).

## Nomogram with multiple clinical features

Univariate COX analysis revealed that age (hazard ratio $= 1.02$, 95% CI [1.01,1.04], $p = 0.005$), T stage (hazard ratio $= 1.73$, 95% CI [1.13,2.65], $p = 0.011$), stage (hazard ratio $= 1.78$ 95% CI [1.25, 2.56], $p = 0.002$), and IAS (hazard ratio $= 2.72$, 95% CI [2.1, 3.51], $p < 0.001$) were independent prognostic predictors for GC, and IAS exhibited improved predictive performance than the three conventional predictors (Fig. 11A). Multivariate COX analysis was performed on age, T stage, stage, and IAS, and age (hazard ratio $= 1.02$, 95% CI [1.01,1.04], $p = 0.007$), stage (hazard ratio $= 1.77$, 95% CI [1.19,2.63], $p = 0.005$), and IAS (hazard ratio $= 2.65$, 95% CI [2.03,3.45], $p < 0.0001$) was independent prognostic predictors of GC (Fig. 11B). Overall, IAS performed better than traditional clinical factors in the prognostic assessment of GC. We further developed a nomogram to assess 1-year,

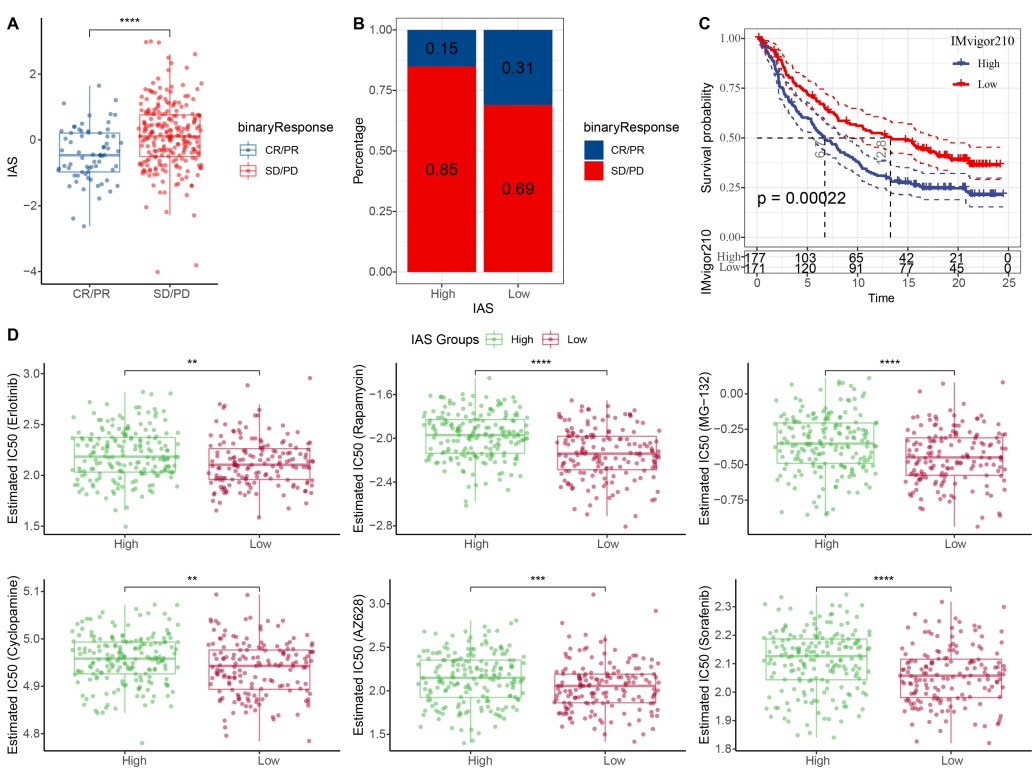

**Figure 9 Treatment prediction for gastric cancer patients.** (A–B) IAS statistics in the IMvigor210 cohort. (C) Kaplan–Meier curves for the IAS groups in the IMvigor210 cohort. (D) IC50 of six drugs in the high and low IAS groups.

2-year, and 3-year survival of GC patients according to Age, Stage, and IAS data (Fig. 11C). In addition, survival prediction for 1, 2, and 3 year(s) was analyzed and the nomogram demonstrated excellent prediction results (Fig. 11D). Decision curve and ROC curve both revealed that compared to conventional clinical features, age, T stage, and stage, the nomogram and IAS exhibited noticeably high accuracy and robustness in prediction (Figs. 11E–11F).

# DISCUSSION

Exosomes secreted by tumor cells could activate CD 8+ T cells and promote their differentiation (*Yao et al., 2013*). CD 8+ T cells are centrally located subpopulation of tumor-killing cells in some solid tumors (*Henning, Roychoudhuri & Restifo, 2018*). The present study showed that cancer immune cycle characterized by higher overall activity score was activated in tumor specimens, showing a remarkable positive correlation with CD 8+ T cells in TME of GC. In contrast, *Zhou et al. (2022)* found that deregulation of CD 8+ T cell glycolysis inhibition in GC mice slows down their depletion, promotes CD 8+ T cell infiltration in TME and further enhances the tumor-killing effect of anti-PD-1 drugs. *Li et al. (2022)* found no typical depleted CD 8+ T cell clusters in GC in their immune cell single cell sequencing analysis. The present study performed WGCNA to mine the gene

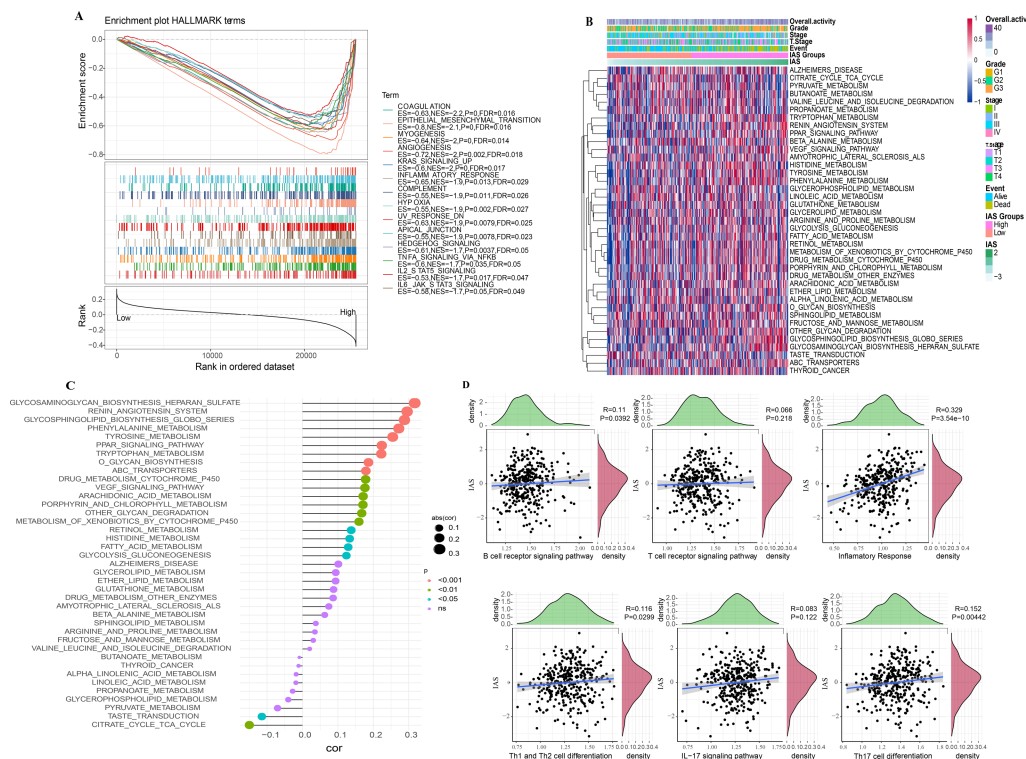

**Figure 10 SsGSEA and GSEA.** (A) GSEA results. (B) ssGSEA results. (C) Spearman correlation bar graph of immune pathway activity and IAS. (D) Spearman correlation scatter plot of inflammation-related pathways and IAS.

modules associated with overall activity score in GC. According to the current results of functional analysis, the genes were mainly associated with B cells and T cells response. In a study in hepatocellular carcinoma, increased B cells and T cells infiltration promote the formation of hyperimmune subtypes as a regulatory factor (*Kurebayashi et al., 2018*). Our results further confirmed this finding. We observed that most immune-related pathways were positively correlated with the overall activity score, which suggested that the overall activity score could serve as an effective indicator of TME activity in gastric cancer.

Seven immune signature genes (AKAP5, CTLA4, LRRC8C, AOAH-IT1, NPC2, RGS1, and SLC2A3) were used to develop the IAS for GC prognosis estimation. AKAP5 is high-expressed in non-mucin producing stomach adenocarcinoma (NMSA) and might modulate gastric carcinogenesis *via* the estrogen signaling pathway (*Zhong et al., 2020*).

However, in our study, the expression level of AKAP5 was downregulated in HGC-27 and AGS cells. Other types such as cystic, mucinous and plasmacytoid tumors in TCGA were excluded by *Zhong et al. (2020)* Common bulk transcriptome analysis was performed by extracting tissues for sequencing analysis. As GC is a highly heterogeneous cancer, individual differences can lead to differences in gene expression levels at cellular level (*Smyth et al., 2020*), which might explain varied results in different studies. Detecting AKAP5 in GC cells at single-cell level seemed to be accurate and reasonable. This study found that AKAP5 as a

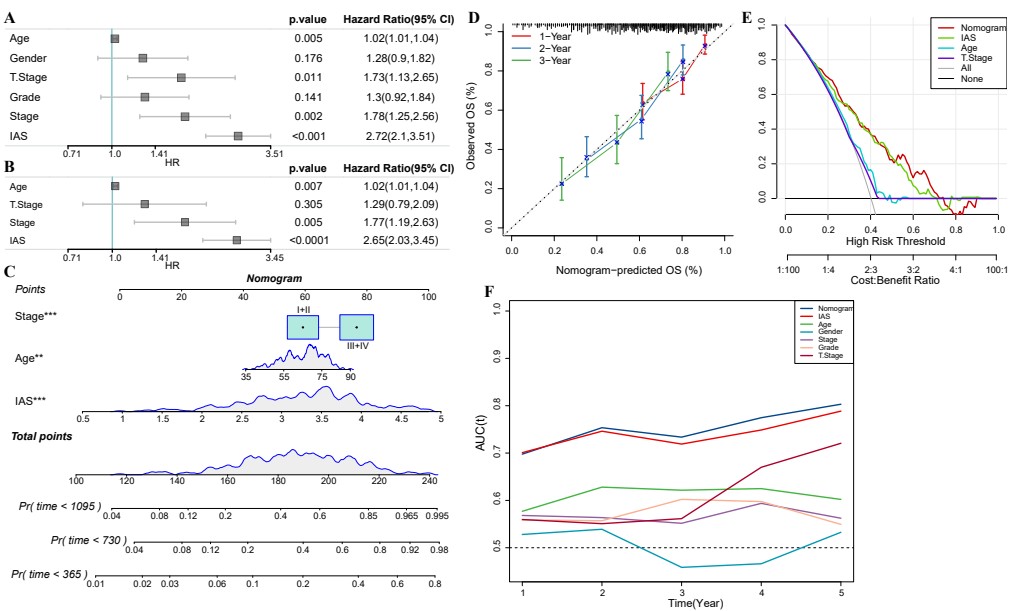

**Figure 11  Nomogram with multiple clinical features.** (A–B) Forest plot of univariate and multivariate COX results of clinical information. (C) Nomogram. (D) Calibration curve. (E) Decision curve. (F) ROC curve of clinical factors, IAS, and nomogram.

protective factor in GC patients showed higher expression level in patients with low IAS, but *Zhong et al. (2020)* pointed out that AKAP5 is a protective factor when it is low-expressed. Different subgroups or threshold settings may lead to different results during COX analysis (*Deng et al., 2017*). CTLA4 is specifically expressed in activated T cells, regulating T cell activation activity at an early stage and acting as an essential regulator of autoimmune defense, while suppressed expression in T cells substantially reduces autoimmune and antitumor activities (*Shiravand et al., 2022*). LRRC8C is a potential cancer gene-related gene, and most of the current studies focused on the immune system. *Concepcion et al. (2022)* showed that LRRC8C mediates $2'3'$cGAMP translocation in T cells, leading to STING and p53 activation, which in turn inhibits T cell function. NPC2 facilitates cytosolic lipid droplet catabolism to maintain macrophage homeostasis (*Robichaud et al., 2021*). RGS1 is a pro-oncogene contributing to osteosarcoma development, whereas miR-376b-3p inhibits osteosarcoma cell proliferation, metastasis and apoptosis by suppressing RGS1 function (*Zhang et al., 2021*). SLC2A3 is an immune biomarker of macrophage infiltration in GC, and SLC2A3 activates aerobic glycolysis in GC cells. SLC2A3-STAT3-SLC2A3 axis activates the downstream STAT3 pathway to promote glycolytic gene phosphorylation, thereby increasing macrophage M2 polarization (*Yao et al., 2020*). In this study, AKAP5 and SLC2A3 in a 7-gene signature were relevant to the mechanism of immune cell infiltration in GC.AOAH-IT1 is a recently identified molecular marker of tumor, and CTLA4, LRRC8C, AOAH-IT1, NPC2, and RGS1 are involved in immune response and serve as molecular markers for therapeutic targets, prognosis, and immune cell immune infiltration in cancer.

Treatment response is a primary prerequisite for improving cancer survival. Although immunotherapy is a promising treatment approach available for some patients with advanced cancer, there is no remarkable therapeutic effect available to GC. A phase III KEYNOTE clinical study indicated that the anti-PD-1 drug pembrolizumab is not therapeutically satisfactory in patients with advanced GC, and pembrolizumab could not measurably improve patients' overall survival when compared to combination chemotherapy or using chemotherapy alone (*Shitara et al., 2020*). Therefore, developing gene signatures for indicating immunotherapy response is also imminently needed. Interestingly, our results illustrated that IAS not only functioned as a predictor of immunotherapy response in GC patients, but also guided chemotherapy drug selection. High abundance of CD8 T cell, Dendritic cell, and NK cell infiltration was observed in patients in the high IAS group. Recruitment of these cells would form an inflammatory TME to promote immunotherapeutic response (*Gajewski et al., 2017*). However, TIDE results suggested that patients in the high IAS group were not amenable to immunotherapy. *Li et al. (2023)* pointed out that the mutation frequency of TTN, TP53, and TMB affects the immunotherapy effect in GC (*Li et al., 2023*). High TMB is favorable for immunotherapy effect (*Chan et al., 2019*). In our results, IAS and TMB showed negative correlation. In addition, TIDE also demonstrated negative correlation with immunotherapy response, and patients with high IAS had the highest TIDE scores, which may explain unsatisfactory immunotherapy results in those patients. The IAS might be a promising evaluation system for reflecting the prognosis of GC patients and precision medicine in the future.

We found that some cancer-related pathways were significantly activated in the high IAS group, for example, ANGIOGENESIS, IL2_STAT5_SIGNALING, IL6_JAK_STAT3_SIGNALING. Angiogenesis factors are overexpressed in cancer progression (*Viallard & Larrivee, 2017*). Activated JAK-STAT pathway leads to GC cell proliferation and tumor progression (*Wang et al., 2020*). It was found that miR-515-5p inhibits hepatocellular carcinoma progression by inhibiting IL6/JAK/STAT3 (*Ni et al., 2020*). From the results, it was observed that these cancer-related pathways were activated in the high IAS group, which may contribute to their poor survival outcomes. Elevated IAS resulted in enhanced metabolic functional response activity in GC. Previous researchers pointed out that depletion of metabolic substances in tumor cells to inhibit cellular metabolic responses is a novel approach to inhibiting GC progression (*Chen et al., 2023*). In summary, the biological pathways enriched in the high IAS group in this study were all positively associated with cancer progression, which led to different prognostic performances.

Limitations of this research should not be neglected. Firstly, the role of five of the seven genes in GC was less analyzed, and their specific molecular mechanisms require further *in vivo/in vitro* assays. Secondly, due to the absence of immunotherapy sequencing data in GC patients, we explored the performance of IAS in predicting immunotherapy response using a cohort of immunotherapy patients with metastatic uroepithelial carcinoma, and further data collection on IAS in GC are needed for validation. To conclude, this study developed a successful evaluation system for indicating immune activity in GC based on the TCGA database. The IAS showed excellent performance in predicting prognosis, immune activity

status, immunotherapy response, and chemotherapy drug resistance in GC. Our study provided novel insights into prognostic assessment in GC.

## Abbreviations

| | |
|---|---|
| **TIP** | Tracking Tumor Immunophenotype |
| **WGCNA** | weighted gene co-expression network analysis |
| **TCGA** | The Cancer Genome Atlas |
| **IAS** | immune activity score |
| **TME** | tumor microenvironment |
| **ICI** | Immune checkpoint inhibitors |
| **ICB** | Immune checkpoint blockade |
| **SNV** | single nucleotide variants |
| **CNV** | copy number variatiosn |
| **GEO** | Gene Expression Omnibus |
| **DEGs** | differentially expressed genes |
| **GO** | Gene Ontology |
| **KEGG** | Kyoto Encylopaedia of Genes and Genomes |
| **LASSO** | Least absolute shrinkage and selection operator |
| **TMB** | tumor mutation burden |
| **CR** | complete response |
| **PR** | partial response |
| **SD** | stable disease |
| **PD** | progressive disease |
| **GDSC** | Genomics of Drug Sensitivity in Cancer database |
| **IC50** | half maximal inhibitory concentration |

### Funding

The research was supported by the Basic Research Program of Shenzhen Innovation Council (JCYJ20210324105609024), the Shenzhen Science and Technology Innovation Commission Project (JCYJ20190809100005672), the Shenzhen Sanming Project (SZSM201612041), Shenzhen Science, Technology Innovation Commission Project (ZDSYS201909020 92855097), the Guangzhou Science and Technology Project of Traditional Chinese medicine and Integrated Traditional and Western Medicine (NO. 803019534036). The funders had no role in study design, data collection and analysis, decision to publish, or preparation of the manuscript.

### Grant Disclosures

The following grant information was disclosed by the authors:
Basic Research Program of Shenzhen Innovation Council: JCYJ20210324105609024.
Shenzhen Science and Technology Innovation Commission Project: JCYJ20190809100005672.
Shenzhen Sanming Project: SZSM201612041.
Shenzhen Science, Technology Innovation Commission Project: ZDSYS201909020 92855097.
Guangzhou Science and Technology Project of Traditional Chinese medicine and Integrated Traditional and Western Medicine: 803019534036.

## Competing Interests

The authors declare there are no competing interests.

## Author Contributions

- Xuan Wu conceived and designed the experiments, performed the experiments, analyzed the data, authored or reviewed drafts of the article, and approved the final draft.
- Fengrui Zhou conceived and designed the experiments, performed the experiments, analyzed the data, authored or reviewed drafts of the article, and approved the final draft.
- Boran Cheng conceived and designed the experiments, performed the experiments, authored or reviewed drafts of the article, and approved the final draft.
- Gangling Tong performed the experiments, authored or reviewed drafts of the article, and approved the final draft.
- Minhua Chen analyzed the data, prepared figures and/or tables, and approved the final draft.
- Lirui He analyzed the data, authored or reviewed drafts of the article, and approved the final draft.
- Zhu Li conceived and designed the experiments, performed the experiments, analyzed the data, prepared figures and/or tables, authored or reviewed drafts of the article, and approved the final draft.
- Shaokang Yu performed the experiments, analyzed the data, prepared figures and/or tables, and approved the final draft.
- Shubin Wang analyzed the data, authored or reviewed drafts of the article, and approved the final draft.
- Liping Lin conceived and designed the experiments, prepared figures and/or tables, authored or reviewed drafts of the article, and approved the final draft.

## Data Availability

The data is available at NCBI GEO: GSE26942 and GitHub: https://github.com/12Xuxuan/Raw-data.git.

## Supplemental Information

Supplemental information for this article can be found online at http://dx.doi.org/10.7717/peerj.16317#supplemental-information.

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
