# Peer review of "Immune activity score to assess the prognosis, immunotherapy and chemotherapy response in gastric cancer and experimental validation"

_PeerJ, doi:10.7717/peerj.16317_

## Round 0.1 · original submission · Major Revisions

Thank you so much for letting us evaluate your work. Although the two reviewers expressed interest in your work, they also made a series of revisions or suggestions. I hope you can take them seriously and reply one by one.

**Language Note:** The review process has identified that the English language must be improved. PeerJ can provide language editing services - please contact us at copyediting@peerj.com for pricing (be sure to provide your manuscript number and title). Alternatively, you should make your own arrangements to improve the language quality and provide details in your response letter. – PeerJ Staff

Reviewer 1 ·

Basic reporting

In this study,authors established an Immune activity score (IAS) model through a variety of bioinformatics analysis to assess prognosis, immunotherapy and chemotherapy response in gastric cancer patients. 7 Hub genes were also well validated. Overall, it meets the publication standards. However, there are still several issues that require attention and improvement before publication.
1. There are spelling errors, inconsistent spelling and semantic duplication in the article. Authors need to revise the paper carefully.
Inconsistent spelling: such as “Overall Activity score”, “Overall activity score” “Overall. activity Score” and Overall activity Score (Figure 3D); 7-step” and “7-steps”

2. In line 241-245, the long sentence should split into several sentence. "

3. The sentence “Both cell lines” should be changed into “All cell lines”
The sentence “gastric cancer patients with low IAS enjoyed high survivorship and long survival, in addition the expression pattern of 7-gene was demonstrated in Figure 4D” should be changed into “gastric cancer patients with low IAS enjoyed high survivorship and long survival. In addition, the expression pattern of 7-gene was demonstrated in Figure 4D”
The sentence “Compared to the high IAS group, the IC50 for Erlotinib (p<0.01), Rapamycin (p<0.0001), MG-132 (p<0.0001), Cyclopamine (p<0.01), AZ628 (p<0.001), and Sorafenib (p< 0.0001) exhibited higher IC50 values” should be changed into “Compared to the high IAS group, the IC50 for Erlotinib (p<0.01), Rap should be changed into amycin (p<0.0001), MG-132 (p<0.0001), Cyclopamine (p<0.01), AZ628 (p<0.001), and Sorafenib (p< 0.0001) was higher”
The sentence “SNV and CNV mutation data” should be changed into “SNV mutation and CNV data”
The sentence “14 pathways existing in the HALLMARK database” should be changed into “14 HALLMARK pathways in the MsigDB database”
The Figure legend for 6C should be changed into “Heat map describing the frequency of CNV events between high-risk and low-risk groups.” The figure legend for 6D should be changed into “The top 15 genes with the highest mutation frequency between the high-risk and low-risk groups”

4. “IAS is a dependable prognostic indicator”? what does it mean?
“frontal mutation frequencies higher than 5%”? what does it mean?

5. In Figure 1, Figure legend does not match the figure

6. The meanings of (*) and (ns) in Figures 1, 6, 8, and 9, and the meanings of (X) in Figures 2C and 8C? Authors need to indicate.

7. Font size in Figures is too small to see, the author(s) should adjust the Font with suitable size. Especially in Figure 1, 2, 6, 8 and 10.

8. “These genes were enriched in 51 235 KEGG pathways, 900 GO terms (BP: 755, CC: 76, MF: 69), containing T cell responses, B cell 236 responses” Actually, only ten pathways/ GO terms were displayed in the Figure 3E, please change relevant description in the text.

Experimental design

1. AKAP5 was up-regulated in non-mucin producing stomach adenocarcinoma (NMSA) in the literature “PMID: 32175408” you mentioned. However, in your study, AKAP5 was identified as a protective factor in univariate COX results. In your in vitro validation experiment, AKAP5 was truly decreased in HGC-27 and AGS cells comparing with RGM-1 cells. You need to explain the difference in discussion part. In addition, the author carried out cell viability. In order to well describe the function of AKAP5, I think overexpressed AKAP5 carrier but not si AKAP5 carrier should be used for cell viability.

2. In the result “Correlation of IAS with clinical features and mutational features”, the descriptions about “mutational features” should be located before the descriptions concerning “clinical features”.
The content of " GSEA and ssGSEA " should be located before the content of " Nomogram with multiple clinical features ", as it is more reasonable to conduct molecular feature analysis before clinical application.

3. From results “Association among IAS and TME scores and gastric cancer treatment” IAS was positively associated with StromalScore, ImmuneScore, ESTIMATEScore and 19 immune cells immunoreactivity, it indicates high degree of immune infiltration in high IAS group. Authors need to explain the finding with TIDE in terms of immune infiltration and Immunotherapy possibilities

4. There is a lack of discussion on the analysis of GSEA and ssGSEA results. It is recommended to supplement it in the discussion section.

5. Authors analyzed variations in the tumor genome in two IAS groups,as displayed in Figure 6C,D. I wonder if there were difference in SNV mutation and SNV data between high and low IAS groups. Chi-squared test should be performed for statistical difference analysis.

6. Why do authors only select these 6 drugs for drug sensitivity testing? In addition,Is IC50 a suitable single indicator to determine drug sensitivity? What is the significance of selecting IC50 data for analysis? And provide detailed calculation methods or citations.

7. In the discussion part,authors say ”The present study showed that cancer immune cycle was activated in tumor specimens, they were characterized by higher overall activity score, which exhibited remarkable positive correlation with CD 8+ T cells in TME of gastric cancer”. However, decreased CD 8+ T cells were discovered in other’s research (Zhou et al. 2022 and Li et al. 2022). Authors should further explain the possible reason for the different findings in the discussion part.

8. CTLA-4 is one of the inhibitory receptors expressed on the surface of T cells, which negatively regulates T cell-mediated immune responses; Tumor cells utilize these inhibitory molecules to induce tumor tolerance and T cell depletion (PMID: 33277742). CTLA-4 is an important finding, which could connect immune activity score and immunotherapy in gastric cancer better. Authors should briefly describe the research progress in CTLA-4 such as immune checkpoint inhibitors (ICIs) and targeting drugs.

Validity of the findings

No comment

Additional comments

No comment

Reviewer 2 ·

Basic reporting

Gastric cancer is extremely heterogeneous, mainly in terms of its heterogeneous tumor microenvironment, which may be associated with differential patient prognosis. In this study, based on the Tracking Tumor Immunophenotype (TIP), the immunoactivity score (IAS) was created and its predictive value in the prognosis of gastric cancer patients was evaluated by means of WGCNA and other bioinformatic analyses. In particular, markers related to the IAS and the prognosis were established, and their expression levels and effects on the proliferation of cancer cells were verified by cellular experiments. In addition, the association of these markers with patients' immunotherapy response and chemotherapy response was established. Overall, this is a study of bioinformatic analysis combined with cellular experiments, and the overall idea is rigorous and logical.

Experimental design

This thesis as a whole used a common set of bioinformatics tools and performed some in vitro confirmations. The following experiments and details are suggested to be added in order to make the results of this study more informative:
1. The association of IAS with immunotherapy is a key finding of the study, and it is recommended that vitro experiments related to immune checkpoint inhibitors be added to further support this finding.
2. AOAH-IT1 and AKAP5, as significant genes tapped in this study, it is recommended to add Transwell or scratch healing experiments in addition to the CCK-8 experiments that have been completed in this study to further demonstrate their effects on cancer progression. At the same time, it is recommended to add experiments about the relative expression levels of the two genes in tumor tissues if conditions permit.
3. Check the experimental data in Figure 5 to ensure the accuracy, and add significance analysis for each time point in Figure 5H-K, labeled at each time point.
4. It is recommended to establish a PPI network of the seven characterized genes with the existing immune-related genes.

Validity of the findings

1. Whether the screening from 16 genes to 7 genes in rows 240-247 passed Lasso and multi-COX analysis, please add the description.
2. The analysis of the correlation between the IAS and the immunotherapy response in this study is insufficient, so we suggest to add the correlation analysis between IAS and the expression of immune checkpoint genes, and add the results in Figure 8.

Additional comments

There are still some flaws that need to be added and improved:
1. In the introduction, it is suggested to add the description of existing studies on mining cancer-related markers based on TIP, so as to make the purpose of this study more clear.
2. It is recommended to add references to some experimental methods to help readers better understand them, such as TIDE-based drug mining in lines 156-163.
3. In the Discussion section, it is suggested to add the research dimension about the molecular mechanism of the 5 genes, such as cytokine release, cell infiltration, and so on.
4. the description of the 7 genes in the discussion, it is recommended not to pile up the existing studies, but to combine these studies with the experimental results of this study and add appropriate speculation.
5. It is suggested to add the abbreviation of gastric cancer in the abstract section and to use the abbreviation to indicate the cancer later.

---

## Round 0.2 · accepted · Accept

The revisions you made have significantly improved the clarity, organization, and overall quality of the manuscript. The reviewers were highly satisfied with the modifications and recognized the merit of your research. I would like to congratulate you on your excellent work and for addressing the reviewers' comments in a thoughtful and comprehensive manner. However, I encourage you to carefully proofread the final version of your manuscript, ensuring that all changes have been accurately incorporated. Furthermore, please adhere to our journal's formatting guidelines and submit the final version as soon as possible.

Reviewer 1 ·

Basic reporting

no comment

Experimental design

no comment

Validity of the findings

no comment

Additional comments

Thank you for the author's careful response, which basically solved my problem.

Reviewer 2 ·

Basic reporting

no comment

Experimental design

no comment

Validity of the findings

no comment

Additional comments

The author provided a comprehensive and clear response to the concerns of the reviewers; Now they are more readable and meaningful.